# Coexistence of SOS-Dependent and SOS-Independent Regulation of DNA Repair Genes in Radiation-Resistant *Deinococcus* Bacteria

**DOI:** 10.3390/cells10040924

**Published:** 2021-04-16

**Authors:** Laurence Blanchard, Arjan de Groot

**Affiliations:** Molecular and Environmental Microbiology Team, Aix Marseille Univ, CEA, CNRS, BIAM, F-13108 Saint Paul-Lez-Durance, France; laurence.blanchard@cea.fr

**Keywords:** *Deinococcus*, radiation resistance, DNA repair, SOS response, SOS mutagenesis, translesion polymerase, DnaE2, SOS-independent, metallopeptidase IrrE, repressor DdrO

## Abstract

*Deinococcus* bacteria are extremely resistant to radiation and able to repair a shattered genome in an essentially error-free manner after exposure to high doses of radiation or prolonged desiccation. An efficient, SOS-independent response mechanism to induce various DNA repair genes such as *recA* is essential for radiation resistance. This pathway, called radiation/desiccation response, is controlled by metallopeptidase IrrE and repressor DdrO that are highly conserved in *Deinococcus*. Among various *Deinococcus* species, *Deinococcus radiodurans* has been studied most extensively. Its genome encodes classical DNA repair proteins for error-free repair but no error-prone translesion DNA polymerases, which may suggest that absence of mutagenic lesion bypass is crucial for error-free repair of massive DNA damage. However, many other radiation-resistant *Deinococcus* species do possess translesion polymerases, and radiation-induced mutagenesis has been demonstrated. At least dozens of *Deinococcus* species contain a mutagenesis cassette, and some even two cassettes, encoding error-prone translesion polymerase DnaE2 and two other proteins, ImuY and ImuB-C, that are probable accessory factors required for DnaE2 activity. Expression of this mutagenesis cassette is under control of the SOS regulators RecA and LexA. In this paper, we review both the RecA/LexA-controlled mutagenesis and the IrrE/DdrO-controlled radiation/desiccation response in *Deinococcus*.

## 1. Introduction

*Deinococcus* bacteria are famous for their extreme resistance to high doses of ionizing radiation and other oxidative stress- and DNA damage-inducing conditions such as desiccation, and for their capacity to repair massive DNA damage including hundreds of double-strand breaks [1,2,3,4,5,6,7]. Many studies, performed with *Deinococcus radiodurans* (the type species of the genus) and also with various other *Deinococcus* species, have indicated that several mechanisms and many proteins contribute to the extreme resistance [5,7]. One important mechanism is the limitation of Fe^2+^-catalysed oxidative damage to proteins due to a high intracellular Mn^2+^/Fe^2+^ ratio and the accumulation of small antioxidant complexes containing Mn^2+^, phosphate and peptides [8,9,10]. Miroslav Radman’s team showed that cell death following exposure to gamma or UV radiation is caused by protein damage in both *D. radiodurans* and radiation-sensitive *Escherichia coli*, but much higher radiation doses are required to generate lethal protein damage in *Deinococcus* [11]. The limitation of protein damage allows the preservation of essential processes including transcription, translation and DNA repair [6,12]. *Deinococcus* bacteria possess homologs of well-known DNA repair proteins (e.g., RecA and UvrABC) involved in classical repair mechanisms such as homologous recombination and nucleotide excision repair [7,13]. Several proteins more specific to *Deinococcus* contribute to DNA repair (e.g., the single-stranded DNA binding proteins DdrA and DdrB) [14,15,16]. The mechanism for repair of hundreds of double-strand breaks in *D. radiodurans* was elucidated by Radman and colleagues [17,18]. In this mechanism, called extended synthesis-dependent strand annealing (ESDSA), radiation-induced DNA fragments with overlapping homologies are used both as primers and as templates for massive synthesis of complementary single strands that allow assembly of genomic fragments through annealing. The expression of *recA*, *uvrB*, *ddrB* and several other DNA repair genes is induced after exposure of *Deinococcus* to radiation or desiccation [19,20,21]. The radiation/desiccation response mechanism to induce these genes is essential for radiation resistance, because mutations abolishing this response are extremely sensitive to radiation [22,23,24,25]. Many bacteria use the RecA/LexA-controlled SOS response to induce DNA repair genes, where activated RecA stimulates the autocleavage of LexA, the repressor of the SOS regulon [26,27]. In *Deinococcus*, induction of *recA* and other DNA repair genes occurs in an SOS-independent manner and is under control of two proteins called IrrE and DdrO [28]. Interestingly, several *Deinococcus* species possess not only the IrrE/DdrO system but also an SOS-dependent pathway that controls expression of a subset of DNA repair genes. Here, after concisely describing the SOS response in *E. coli* and some other bacteria, we review the SOS-dependent and SOS-independent responses in *Deinococcus*.

## 2. SOS Response and SOS-Induced Mutagenesis in *E. coli* and Many Other Bacteria

### 2.1. Escherichia coli

The SOS response, discovered and named by Miroslav Radman [29] has been studied in many bacteria, in particular in the model bacterium *E. coli* [27,30]. The molecular signal that induces the SOS response is the presence of single-stranded DNA (ssDNA) that accumulates when DNA damage occurs. ATP-bound RecA filaments formed on ssDNA stimulate the self-cleavage of LexA, the transcriptional repressor of the SOS system. Intact LexA binds as a dimer to specific sequences in the operator regions of DNA repair and other genes that constitute the SOS regulon. The RecA filaments induce cleavage of LexA that is unbound to DNA, leading to a decrease in the unbound LexA pool provoking dissociation of LexA from its target DNA sites and de-repression of the SOS regulon genes [31]. RecA-mediated cleavage of LexA has been reconstituted in vitro, and requires the presence of ssDNA, ATP (or a non-hydrolysable analogue) and Mg^2+^ [32,33]. Even though the SOS response mechanism is widespread, the number and type of genes of the SOS regulon vary considerably among bacteria [26].

The SOS regulon in *E. coli* comprises about 40 genes, encoding, amongst others, DNA repair proteins involved in homologous recombination (e.g., RecA), nucleotide excision repair (e.g., UvrA) and error-prone DNA translesion synthesis (DNA polymerases Pol II, Pol IV and Pol V). Pol II is a B-family translesion synthesis (TLS) polymerase, and Pol IV and Pol V are Y-family TLS polymerases. Pol V, the product of the *umuDC* genes, is responsible for most SOS-induced mutagenesis in *E. coli* [34]. Pol V is a trimer composed of UmuC and a dimer of UmuD’. The latter is produced after UmuD autocleavage, which, similar to LexA, is stimulated by RecA nucleofilaments. RecA is not only required for Pol V induction and formation but also for Pol V activity [35,36,37]. In addition to LexA and UmuD cleavage, RecA nucleofilaments also catalyse autocleavage of phage lambda repressor and related phage repressors, inducing prophage excision and bacterial cell lysis. Furthermore, although the SOS response is particularly known for induction of DNA repair genes and SOS mutagenesis, SOS-induced dormancy (mediated by the toxin of one or more toxin-antitoxin systems) as well as LexA-regulated programmed cell death have also been described in some bacteria [27,31,38,39,40].

### 2.2. The Widespread imuA-imuB-dnaE2 Mutagenesis Cassette

Instead of Pol V, many different bacteria possess another SOS-controlled mutagenesis cassette consisting of *imuA*, *imuB* and *dnaE2* [36,41,42]. Several species possess these genes in a single operon together with a *lexA* homolog (Figure 1), whereas this mutagenesis cassette is split in other bacteria (*lexA* separated from *imuAB-dnaE2*, or *lexA*, *imuAB* and *dnaE2* at three or four separate locations on the genome) [26]. SOS mutagenesis by this cassette has been studied in particular in *Caulobacter crescentus* (*Caulobacter vibrioides*) [41,43], *Mycobacterium tuberculosis* [44,45] and *Pseudomonas* spp. [46,47,48]. ImuA and ImuB are accessory factors required for the error-prone TLS performed by DnaE2 (also called ImuC), a paralog of the α subunit DnaE of the bacterial replicative polymerase. ImuA has some similarity with RecA. ImuB contains an Y-family polymerase domain, but it is catalytically dead because it lacks conserved active site residues required for polymerase activity [45]. ImuB proteins also have a conserved C-terminal region called ImuB-C, and a bioinformatics study has found that ImuB-C also occurs as a standalone protein in various bacteria [49]. The precise role of the ImuB-C domain or standalone protein is not clear, but it has been proposed that it might bind to damaged DNA and/or might interact with DnaE2 [49,50]. It should be noted, however, that the gene encoding standalone ImuB-C is not only found adjacent to *dnaE2* but also next to other DNA repair genes encoding, for example, Pol IV in various bacteria [49]. Concerning SOS mutagenesis, experiments with *C. crescentus* have shown that RecA is only required for induction of the *imuAB-dnaE2* genes but not for the TLS mediated by ImuAB-DnaE2 [43]. Since RecA is required for Pol V to promote lesion bypass in *E. coli*, it has been suggested that ImuA might function analogously to RecA for ImuAB-DnaE2-mediated TLS [43,50].

## 3. SOS Response and SOS-Induced Mutagenesis in *Deinococcus* Bacteria

### 3.1. Absence of SOS Response in Deinococcus radiodurans

*D. radiodurans* and *Deinococcus geothermalis* were the first two *Deinococcus* species for which the complete genome sequence was obtained and annotated [13,51,52]. Both genomes showed the presence of classical DNA repair genes involved in error-free repair pathways such as homologous recombination, base excision repair and nucleotide excision repair. Genes encoding TLS polymerases were not found, in line with the idea that error-prone activity of TLS polymerases might be disadvantageous in case of thousands of DNA lesions generated by high doses of radiation or prolonged desiccation [53]. Following exposure to radiation or desiccation, *recA* and several other DNA repair genes were found to be induced in *D. radiodurans* [19,20], indicating the presence of one or more regulatory mechanisms to respond to DNA damaging conditions. Indeed, *D. radiodurans* possesses two LexA homologs, both of which are cleaved in a RecA-dependent manner in vitro and in vivo. However, these LexA proteins are not involved in controlling the expression of *recA* and other DNA repair genes in *D. radiodurans* [54,55,56,57,58]. Instead, these genes appear to be regulated by an SOS-independent mechanism involving the proteins IrrE and DdrO (see Section 4). The genes regulated by the two *D. radiodurans* LexA proteins are currently unknown. LexA homologs are found in most, but not all, other sequenced *Deinococcus* species [7].

### 3.2. SOS-Induced Mutagenesis in Deinococcus deserti

The third available complete *Deinococcus* genome sequence was that of *Deinococcus deserti* [4]. Remarkably, several additional DNA repair proteins compared to *D. radiodurans* and *D. geothermalis* were found in *D. deserti*, including homologs of three TLS polymerases: Pol II (PolB), ImuY containing an Y-family polymerase domain, and DnaE2 [59]. *D. deserti* also possesses three *recA* genes coding for two different RecA proteins that share 81% identity: RecA_C_ and RecA_P_. The RecA_C_ protein is encoded by the chromosome and RecA_P_ is the identical product encoded by two extra *recA* genes present on two different plasmids. Radiation-induced expression was observed for each *recA* gene and for *imuY* and *dnaE2* but not for *polB* [4,21,59]. The presence of the TLS polymerases suggested that error-prone lesion bypass could occur in *D. deserti*. UV-induced mutagenesis was indeed observed for the wild-type strain. This mutagenesis was abolished in mutant strains lacking either ImuY, DnaE2 or chromosome-encoded RecA_C_, but was unaffected in strains lacking PolB or the plasmid-encoded RecA_P_ [59]. The ImuY and DnaE2 mutant strains did not show a decrease or increase in radiation resistance under the experimental conditions.

Interestingly, the *imuY* and *dnaE2* genes are present in an operon that is similar, but not identical, to the *lexA-imuA-imuB-dnaE2* mutagenesis cassette found in other bacteria (Figure 1) (see also Section 2.2). A homolog of *imuA* was not found in this operon or elsewhere in the *D. deserti* genome. However, another gene encoding a hypothetical protein of 80 amino acid residues is present between *imuY* and *dnaE2*. The name ImuY was given to the Y-polymerase homolog because of the low sequence similarity with ImuB and because *imuA* is absent [59]. The presence of *lexA* in this operon strongly suggested that the operon is SOS regulated. Indeed, the induction of this operon upon radiation is abolished in a *recA*_C_ mutant (but not in a mutant lacking the two *recA*_P_ genes), in line with the induced mutagenesis results. RecA_C_-dependent cleavage of the LexA protein in vitro was also observed [59]. More recently, the 80-residue hypothetical protein was identified as a standalone ImuB-C protein that has similarity with the C-terminal domain of ImuB proteins [49] (see also Section 2.2). Like ImuB, ImuY is predicted to be an inactive polymerase and likely required, probably together with ImuB-C, as an accessory factor for the error-prone translesion activity of DnaE2 [49,50]. It is currently not known if the ImuY-ImuB-C-DnaE2 proteins are sufficient to perform translesion, or, since ImuA is absent, if this process requires the presence of RecA [50].

### 3.3. Presence of imuY-imuB-C-dnaE2 Mutagenesis Cassette in Many Deinococcus Species

An increasing number of complete or draft genome sequences of other *Deinococcus* species has become available in recent years. A comparison of 11 complete *Deinococcus* genomes showed that *D. deserti* is not an exception with respect to the presence of TLS polymerases [7]. Eight of these 11 species possess one or more of the TLS polymerase homologs Pol II (in four species), Pol IV (seven species) or the ImuY-ImuB-C-DnaE2 mutasome (two species: *D. deserti* and *Deinococcus peraridilitoris*).

Like in *D. deserti*, the *imuY, imuB-C* and *dnaE2* genes of *D. peraridilitoris* are located in a probable operon with *lexA* upstream of *imuY* (Figure 1). Our recent BLAST searches and sequence analyses revealed that this *lexA-imuY-imuB-C-dnaE2* operon is present in at least 20 other *Deinococcus* species, including *Deinococcus ficus* [60], *Deinococcus grandis* [61], *Deinoco**ccus koreensis* [62] and *Deinococcus marmoris* [63] for which the genome sequence has been published (Table 1; Appendix A). Remarkably, several of these species, including *D ficus*, *D grandis* and *D. koreensis*, have two different *lexA-imuY-imuB-C-dnaE2* operons (gene products of one operon are not identical to the counterparts of the second operon). These data strongly suggest that SOS mutagenesis is not uncommon in radiation-resistant *Deinococcus* bacteria. In addition to *D. deserti*, SOS mutagenesis dependent on ImuY and DnaE2 has also been shown in *D. ficus* [64]. In this study, UV-induced mutagenesis was decreased in an *imuY* or *dnaE2* mutant, but not completely abolished, possibly because of the presence of a second *lexA-imuY-imuB-C-dnaE2* mutagenesis cassette.

In addition to the mutagenesis cassette, *D. deserti* and *D. peraridilitoris* have in common that they produce two different RecA proteins [7,59]. Based on the experimental data obtained with *D. deserti*, it was suggested that the extra RecA protein could be involved in controlling the induction levels of the mutagenesis cassette, with the extra RecA (RecA_P_ in *D. deserti*) allowing error-free repair of DNA damage but not inducing error-prone lesion bypass [59]. However, only one *recA* is present in the majority of the *Deinococcus* species possessing a *lexA-imuY-imuB-C-dnaE2* cassette.

In conclusion, many *Deinococcus* species possess TLS polymerases, and SOS mutagenesis implicating ImuY, DnaE2 and, probably, ImuB-C has been demonstrated. BLAST searches revealed that homologs of Pol II and Pol IV are also present in dozens of *Deinococcus* species, but further work is needed to determine if these Pol II and Pol IV polymerases have error-prone activity and if they are upregulated after exposure to radiation or other conditions. Error-prone TLS can contribute to adaptation to harsh conditions by generating genetic variability. However, especially for *Deinococcus* bacteria that are able to tolerate and repair massive DNA damage, mutagenic TLS should be limited and tightly controlled to prevent the generation of many deleterious mutations. Limitation of TLS is probably related to a higher abundance and efficiency of proteins involved in error-free DNA repair, which is supported by the observation that the frequency of UV-induced point mutations was much higher in a *D. deserti* mutant in which the SOS-independent induction of error-free repair proteins such as RecA was abolished [59]. Finally, it is important to mention that induced mutagenesis via transposition of insertion sequence (IS) elements has also been described in two *Deinococcus* species. In *D. radiodurans* and *D. geothermalis*, which lack TLS polymerases, transposition of certain IS elements is induced after exposure to radiation or oxidative stress [65,66,67]. Other *Deinococcus* species, including *D. deserti* [4], also contain IS elements, but it is currently unknown if transposition of one or more of these ISs is stimulated after stress exposure.

## 4. SOS-Independent Radiation Response Mechanism in *Deinococcus* Bacteria

### 4.1. Conserved Radiation/Desiccation Response Mechanism Controlled by IrrE and DdrO

A common palindromic motif, called radiation/desiccation response motif (RDRM), was identified in the upstream regions of about 20 genes, including *recA* and several other DNA repair genes, that are highly upregulated after exposure of *D. radiodurans* to gamma radiation or desiccation, which indicated the presence of a radiation/desiccation response (RDR) regulon controlled by a specific transcription factor [52]. The RDRM was also found upstream of essentially the same genes in other *Deinococcus* species [4,52,68]. RNA sequencing experiments in *D. deserti* showed that the RDRM sites are located close to or overlapping with the transcription start sites of these genes, which indicated that the RDRM is a binding site for a transcriptional repressor [21]. Instead of LexA, which is not involved in *recA* induction in *D. radiodurans* (see Section 3.1), two other candidates for the role of regulator of the RDR regulon were found. The first one was identified in two independent studies after characterization of radiation-sensitive mutants of *D. radiodurans* that appeared to have a mutation in a novel gene designated *irrE* (also referred to as *pprI*) [22,23]. Radiation-induced expression of *recA* and several other genes was abolished in *irrE* mutants, which indicated that IrrE could be a positive regulator of RDR genes [22,23,24,59]. The IrrE sequence [22,23] and the crystal structure of *D. deserti* IrrE (Figure 2) [24] suggested that IrrE could have zinc-dependent protease activity. Another study proposed DdrO as the RDR regulator, because *ddrO* is radiation-induced, preceded by an RDRM, and encodes a typical transcriptional regulator containing an N-terminal DNA-binding domain [52]. Important progress in understanding the SOS-independent regulation of RDR genes was obtained by experimentally demonstrating that IrrE and DdrO function together in controlling expression of the RDR regulon [28]. IrrE indeed appeared to have zinc metallopeptidase activity [28,68,69], and is now classified as an M78 family (COG2856) metallo-endopeptidase in the MEROPS Peptidase Database [70]. IrrE cleaves and inactivates DdrO, the transcriptional regulator that binds to the RDRM and thereby functions as repressor of the RDR genes [25,28,68,71,72]. In contrast to *ddrO*, expression of the IrrE gene or protein is not induced after radiation [19,21,73]. DdrO cleavage by IrrE in *Deinococcus* is stimulated after exposure to radiation or other oxidative stress- and DNA damage-generating treatments, resulting in de-repression of the RDR regulon. This proteolytic cleavage is essential for radiation resistance, because strains expressing IrrE with point mutations in the active site or expressing uncleavable DdrO are extremely sensitive to radiation [24,25].

IrrE and DdrO are present and highly conserved in each sequenced *Deinococcus* species [7]. Moreover, both IrrE [24] and DdrO [72] from *D. deserti* can functionally replace their counterparts in *D. radiodurans*, showing conservation of the intracellular molecular mechanisms implicating IrrE and DdrO in *Deinococcus* bacteria. Noteworthy, the translation initiation codon position of many *irrE* genes is wrongly annotated [28,69], and for experimental studies it is of course crucial to take into account the correct start of the gene and protein.

The RDR regulon composition is largely conserved but also shows some variation across *Deinococcus* species [68]. Demonstrated or predicted RDR regulon members include genes encoding the RDR repressor (*ddrO*), DNA repair proteins (*recA*, extra *recA*, *ssb*, *gyrA*, *gyrB*, *recD*, *recQ*, *ruvB*, *uvrA*, *uvrB* and *uvrD*), and proteins more specific to *Deinococcus* and involved in repair or metabolism of DNA (*ddrA*, *ddrB*, *ddrC*, *pprA*) or with unknown function (*ddrD*, *ddrF*, *ddrQ*, *ddrR*, *ddrS*, *ddrTUVWX*). Some of these genes are not present in every *Deinococcus* species (e.g., extra *recA*, *ddrA*, *pprA*). In *D. deserti*, the radiation-induced expression of *recA*_C_ and the *lexA-imuY-imuB-C-dnaE2* mutagenesis cassette requires the presence of *irrE* and *recA*_C_, respectively (see also Section 3.2). However, the mutagenesis cassette is still induced in the *irrE* mutant, indicating that basal level of RecA_C_ is sufficient for this induction [59]. The RDR regulon includes *ddrO* itself, allowing DdrO to re-accumulate and re-repress the RDR regulon when the stress is alleviated. Interestingly, DdrO is essential for cell viability and its prolonged depletion induces apoptotic-like cell death in *D. radiodurans*, suggesting that DdrO, like the SOS repressor LexA in *E. coli* and other bacteria, not only regulates DNA repair genes allowing stress survival but also, directly or indirectly, programmed cell death [71].

### 4.2. IrrE Metallopeptidase Activity and Activation

In addition to the radiation-induced DdrO cleavage in *Deinococcus*, IrrE-dependent DdrO cleavage has also been demonstrated when both proteins are co-expressed from plasmids in *E. coli* without exogenous stress, and in vitro after mixing the two proteins purified from *E. coli* [28]. Addition of zinc ions to metal-free IrrE is sufficient to restore IrrE activity in vitro, and the in vitro cleavage does not require any other specific molecule such as DNA [68]. Besides the metallopeptidase domain, including the zinc-binding active site motif HEXXH, the crystal structure of *D. deserti* IrrE also revealed a C-terminal domain with structural similarity to GAF domains (Figure 2) [24]. GAF domains in other proteins may have a role in signal transduction after binding a small molecule, whereas other GAF domains do not bind a ligand but are involved in protein–protein interactions [74,75]. The possibility that binding of a signal molecule to the GAF-like domain of IrrE has a role in activating the peptidase domain has been considered [24,28]. However, such a molecule is apparently not required for DdrO cleavage in *E. coli* or in vitro. Moreover, although clearly less efficient than with full-length IrrE, inducible DdrO cleavage was still observed with a truncated IrrE protein lacking the GAF-like domain, showing that this domain is not absolutely required for activating IrrE peptidase [69]. As a second hypothesis, it has been proposed that the metallopeptidase activity of IrrE may be regulated by the availability of zinc ions [28]. Free zinc levels in cells are extremely low because nearly all zinc ions are bound to proteins. IrrE is a low abundance protein [4,73] and zinc availability for IrrE may be limited if most of the intracellular zinc is more strongly bound to more abundant proteins or other molecules. Oxidative stress can result in rapid release of zinc ions from cysteine-containing zinc sites after oxidation of the cysteine thiolates that coordinate Zn^2+^, and the released Zn^2+^ can be a potent signal that can be sensed by other proteins [76,77,78]. Oxidative stress-induced zinc release has been reported for several bacterial proteins (e.g., Hsp33) [79,80]. Recent experimental data support this second proposed IrrE activation mechanism. Sudden exposure of *Deinococcus* to zinc excess rapidly induces DdrO cleavage [68], but is not accompanied by production of reactive oxygen species (ROS) and DNA damage, indicating a direct activation of IrrE by increased intracellular levels of free zinc ions [69]. Furthermore, an increase in intracellular free zinc in *Deinococcus* was detected after oxidative treatment. Many proteins with zinc/cysteine sites, and thus possible sources of ROS-induced zinc mobilization, were identified in *D. deserti*. A large fraction of these proteins are involved in DNA replication and repair, and it is tempting to speculate that destruction of their zinc-binding sites inhibits their function, while the mobilized zinc stimulates IrrE activity and expression of DNA repair genes. Together, the data indicate that radiation and oxidative stress induce changes in redox homeostasis, resulting in generation of a zinc signal and subsequent IrrE activation in *Deinococcus* (Figure 3).

### 4.3. Repressor DdrO and IrrE-DdrO Interaction

DdrO from *D. deserti* was studied using a domain dissection approach associated to several experimental methods, including X-ray crystallography [72]. DdrO is composed of two domains that fold independently and are separated by a flexible linker. The N-terminal domain corresponds to the DNA-binding domain of the XRE family. The C-terminal domain is required for DdrO dimerization and for binding of entire DdrO to DNA. Cleavage by IrrE, which occurs between Leu106 and Arg107 in the C-terminal domain of the 129-residue DdrO protein, abolishes dimerization and DNA binding of DdrO. Crystal structures of entire DdrO and the separately produced N- and C-terminal domains were obtained (Figure 2). The N-terminal domain alone is monomeric in solution and in bacterial two-hybrid experiments but, in the crystal, it forms a dimer similar to other XRE family DNA-binding domains and is likely very similar to the dimer that binds the palindromic RDRM. The C-terminal domain alone forms dimers in solution, in bacterial two-hybrid experiments and in the crystal. The dimer structures of the separately produced N- and C-terminal domains are the same as observed in the crystal structure of entire DdrO [72]. The crystal structure of full-length DdrO from *D. geothermalis* has also been solved [81], and is essentially identical to the one of *D. deserti*. Interestingly, the cleavage site is hidden in the dimer, indicating that IrrE rather cleaves DdrO monomer instead of dimer. This is supported by results obtained with DdrO having the D96R mutation in the C-terminal domain [72]. This mutant protein is affected in dimerization (and DNA binding) and is more efficiently cleaved by IrrE in vitro than wild-type DdrO. Cleavage of this D96R mutant protein in vivo in *Deinococcus* requires stress exposure. Furthermore, modelling the IrrE-DdrO interaction using two docking tools resulted in satisfying models, with the cleavage site in DdrO close to the active site in IrrE, only for an IrrE-DdrO monomer complex but not for an IrrE-DdrO dimer complex [72]. The models found in common with the two docking tools also suggest that the GAF-like domain of IrrE may contribute to the interaction with DdrO [69]. In the cells under standard conditions, DdrO exists as a dynamic equilibrium between DNA-bound and free dimers and free monomers (including freshly synthesized monomers). A current model proposes that IrrE-mediated cleavage of unbound DdrO monomers under stress condition will shift this equilibrium toward cleavable monomers, resulting in reduced amount of free DdrO provoking dissociation of DdrO from RDRM sites and thus de-repression of RDR regulon genes (Figure 3). Self-cleavage of the SOS repressor LexA also occurs when dissociated from DNA but not when LexA is bound to target DNA [31]. With cleavage of unbound repressor, the level of repression/de-repression of regulon genes will correlate with the binding affinity of the repressor for the different target sites.

### 4.4. IrrE/DdrO-Related Protein Pairs Widespread in Bacteria

IrrE and DdrO were initially only found in *Deinococcus*. Following the increase in the amount of genome sequences, close homologs of IrrE and DdrO are now also found in several other bacterial species belonging to the *Deinococcus*/*Thermus* phylum, including *Deinobacterium chartae*, *Calidithermus* spp., *Meiothermus* spp., *Marinithermus hydrothermalis*, and *Oceanithermus* spp., but not in *Thermus* spp. [28]. Moreover, COG2856/XRE protein pairs distantly related to IrrE and DdrO are present in many bacteria, often encoded by conjugative transposons and bacteriophages, indicating that stress response mechanisms involving such protein pairs are widespread [82,83]. The stress-induced inactivation of some XRE family repressors by their cognate COG2856 domain-containing antirepressor, inducing prophage or transposon excision, has indeed been demonstrated (e.g., cleavage of ImmR by ImmA in *Bacillus subtilis*) [82,84,85]. Thus, whereas cleavage of certain bacteriophage repressors is stimulated by the activated SOS response component RecA of the host, other mobile genetic elements use IrrE-like proteins for repressor inactivation. As these IrrE-like proteins are predicted or demonstrated metallopeptidases containing the HEXXH motif, but generally lack a GAF-like domain [72], it can be speculated that they are activated by stress-induced increase in zinc ion availability like IrrE in *Deinococcus*.

## 5. Conclusions

Many years of studies with the model bacterium *E. coli* has provided a wealth of insight in many cellular processes including the SOS response. Even though widespread, the SOS system is not uniform across bacterial species and for example shows variation in number and type of regulated genes and in regulatory motif sequences. Similarly, thorough characterization of *D. radiodurans* has led to many important discoveries with respect to the mechanisms underlying its extreme resistance to radiation and repair of massive DNA damage. These mechanisms are probably largely conserved across other members of the genus *Deinococcus*, but interesting diversity has also been observed both experimentally and after comparative analysis of many *Deinococcus* genome sequences. Notably, SOS mutagenesis is absent in *D. radiodurans* and several other *Deinococcus* species, but has been demonstrated in *D. deserti* and *D. ficus* and probably occurs in dozens of other *Deinococcus* bacteria, mediated by a mutagenesis cassette encoding ImuY, ImuB-C and DnaE2. Instead of a classical SOS response mechanism to induce error-free DNA repair, all *Deinococcus* species possess the radiation/desiccation response (RDR) mechanism, controlled by repressor DdrO and the separate metallopeptidase IrrE, to regulate expression of *recA* and several other DNA repair genes.

With respect to the mutagenesis cassette and the RDR system in *Deinococcus*, it will be interesting to elucidate the role of several gene products implicated in or induced by these pathways; for example, the small ImuB-C protein and the proteins of unknown function encoded by RDR regulon genes. Additionally, what could be the advantage, if any, of having two mutagenesis cassettes as observed in several species? The RDR repressor DdrO is essential for viability in at least *D. radiodurans* and *D. deserti*. DdrO depletion in *D. radiodurans* provokes apoptotic-like death, which is a poorly described phenomenon in bacteria and it will thus be very interesting to identify and characterize the genes and proteins implicated in this bacterial programmed death pathway.

## Figures and Tables

**Figure 1 cells-10-00924-f001:**
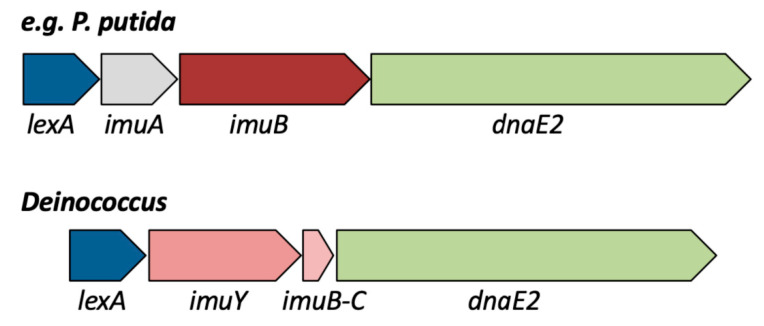
Schematic representation of the *lexA-imuA-imuB-dnaE2* and *lexA-imuY-imuB-C-dnaE2* mutagenesis cassettes. The operon encoding LexA, ImuA, ImuB and DnaE2 is present in bacterial species such as *Pseudomonas putida*, *Pseudomonas fluorescens*, *Pseudomonas syringae*, *Xanthomonas campestris*, *Dechloromonas aromatica*, *Methylococcus capsulatus*, and *Acidithiobacillus ferrooxidans*. The operon encoding LexA, ImuY, ImuB-C and DnaE2 is present in at least 20 *Deinococcus* species, including *D. deserti*, *D. ficus*, *D. peraridilitoris*, *D. marmoris*, *D. koreensis* and *D. grandis*. For *D. deserti*, the locus tags are Deide_1p01870 (*lexA*), Deide_1p01880 (*imuY*), Deide_1p01890 (*imuB-C*) and Deide_1p01900 (*dnaE2*).

**Figure 2 cells-10-00924-f002:**
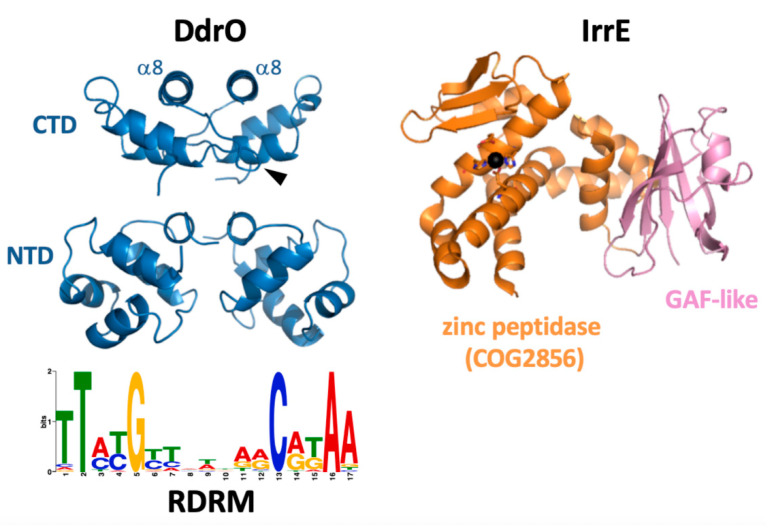
Crystal structures of DdrO and IrrE from *D. deserti*. For DdrO, the dimers of the C-terminal domain (CTD) and the DNA-binding N-terminal domain (NTD) are shown. The DdrO structure contains eight α-helices (five in the NTD, three in the CTD), and IrrE cleaves DdrO in the loop connecting α7 and α8 (one cleavage site is indicated with an arrowhead). Sequence logo of the conserved radiation/desiccation motif (RDRM) to which DdrO binds is also shown.

**Figure 3 cells-10-00924-f003:**
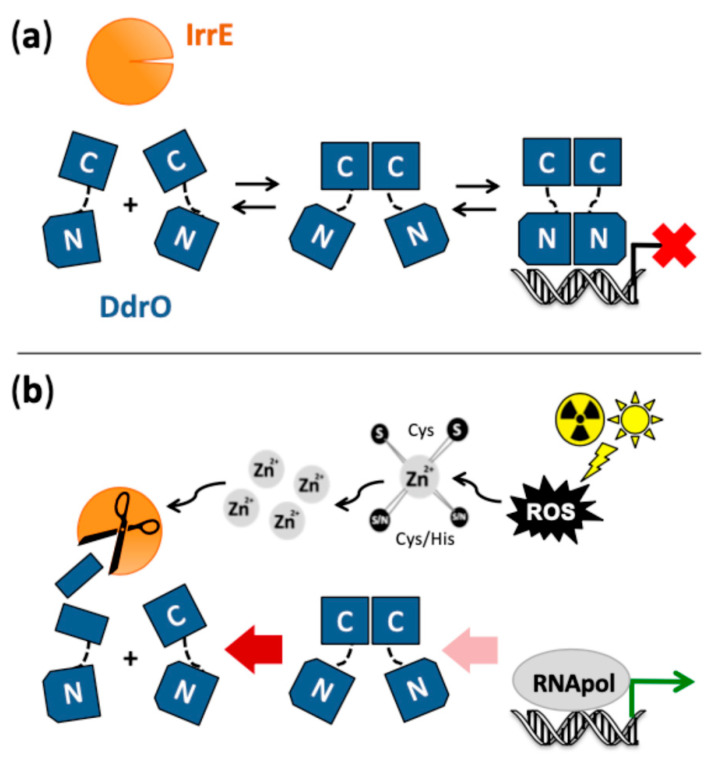
Proposed mechanism for induction of the SOS-independent radiation/desiccation response in *Deinococcus*. (**a**) Under standard conditions, the majority of zinc metallopeptidase IrrE is inactive due to limited availability of zinc ions, and repressor DdrO exists as a dynamic equilibrium between monomers and free and DNA-bound dimers. DdrO contains an N- and C-terminal domain separated by a flexible linker. DdrO dimerization is mediated by its C-terminal domain. Dimers of the N-terminal domain are expected mainly when DdrO is bound to the palindromic target DNA motif located in the promoter region of radiation/desiccation response (RDR) genes, which represses their transcription. (**b**) Exposure to conditions such as radiation and desiccation generates oxidative stress through formation of reactive oxygen species (ROS), which can cause oxidation of cysteine residues of zinc/cysteine sites in proteins, concomitantly causing release of zinc ions from these sites and a transient increase in the intracellular concentration of free, available zinc ions. The released zinc functions as second messenger, and increases the amount of active zinc-bound IrrE that cleaves the C-terminal domain of monomeric DdrO, abolishing its dimerization and shifting the DdrO equilibrium toward cleavable monomers. The diminished amount of DdrO leads to induced expression of the RDR regulon genes (including several DNA repair genes and *ddrO* itself). After the stress is alleviated and the zinc signal has disappeared, zinc availability for (newly synthesized) IrrE will become limited again, allowing DdrO to re-accumulate.

**Table 1 cells-10-00924-t001:** Locus tags of the *lexA-imuY-imuB-C-dnaE2* cassette of complete or published *Deinococcus* genome sequences.

Species	*lexA*	*imuY*	*imuB-C*	*dnaE2*
*D. deserti*	Deide_1p01870	Deide_1p01880	Deide_1p01890	Deide_1p01900
*D. peraridilitoris*	Deipe_2980	Deipe_2981	Deipe_2982	Deipe_2983
*D. ficus*	DFI_00235	DFI_00230	DFI_00225	DFI_00220
*D. ficus*	DFI_19975	DFI_19970	DFI_19965	DFI_19960
*D. grandis*	DEIGR_310081	DEIGR_310080	DEIGR_310079	DEIGR_310077
*D. grandis*	DEIGR_200158	DEIGR_200159	DEIGR_200160	DEIGR_200161
*D. koreensis*	CVO96_07480	CVO96_07485	CVO96_07490	CVO96_07495
*D. koreensis*	CVO96_19955	CVO96_19950	CVO96_19945	CVO96_19940
*D. marmoris*	BOO71_0010528	BOO71_0010525	BOO71_0010522	BOO71_0010519

## Data Availability

IDs of genes and proteins found in this study are available in Table 1 and Appendix A.

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
