# Peer review of "Coexistence of SOS-Dependent and SOS-Independent Regulation of DNA Repair Genes in Radiation-Resistant Deinococcus Bacteria"

_cells, 2021, doi:10.3390/cells10040924_

Round 1
Reviewer 1 Report
This review summarizes current knowledge of the mechanisms that regulate DNA repair genes in Deinococcus spp. The unusual resistance of this bacterial genus to ionizing radiation and to desiccation makes the subject appealing; hence, one may predict wide readership including readers outside the DNA repair community. The authors of the review have done an excellent job combining scientific rigor with readability. The figures are simple and easy to understand. Use of the existing literature is correct. Below I make a number of suggestions, one concerning Fig. 1 and others about the text. None of the suggestions is crucially relevant and some may be superfluous. However, except in poetry the perfect manuscript does not exist, and perhaps my comments may be a little help to polish this excellent review.
In Fig. 1, use of Deinococcus spp. might avoid the need of adding "e. g." in Pseudomonas putida.
L 10 IS "within hours" necessary or superfluous?
L 48 For consistency with a previous sentence that also mentions Radman, I suggest to use past tense (was) instead of present perfect (has been).
L 77 Start the sentence with "RecA-mediated cleavage" (without "The")
L 99. May "demonstrated" be too strong? Might "described" be more prudent?
L 122-123 "several bacteria" may be too informal considering that you mention a number of bacterial species by their names. Suggested text: "is present in bacterial species such as..."
L 137. "Imprudent" transmits well the idea but it is anthropomorphic. I suggest you find another way of saying this or at least a more neutral adjective.
L 141. Is "possess" correct? Shouldn't it be "possesses"?
L 144. For consistency with the previous and the following sentence, use of the present (appears) may be advisable.
L 151. Use of "the one" is colloquial; "that" would be more formal.
L 180. "as has been suggested" can be deleted without loss of clarity.
L 184. As you are speaking of recent years, "has become" may be better than "is becoming".
L 265. Is "intracellular" necessary? May it be superfluous?
L 276. Do you mean in every Deinococcus species? "Each" may be misleading.
L 299-300. Please try to make the sentence "However, at least in E. coli or in vitro such potential (specific) molecule appears not required for DdrO cleavage" less tortuous.
L 327. Footnote of Fig. 3. Suggested text: "Proposed mechanism for induction...".
L 373. "The obtained results" should be replaced with something more formal. How about "A current model proposes..." or something similar?
L 383. "Huge" is informal language. Please find another adjective.
L 402. If you say that the SOS system is "widespread", it is not necessary that you add that "it can even be absent" in certain species. Widespread already indicates that is frequently but not universally found.
Reviewer 2 Report
The Deinococcus species are known for their ability to resist huge doses of radiation or exposure to other damaging agents, whereas small doses dramatically decrease the viability of most of the other bacterial species. The authors submitted a very interesting and original review describing in several Deinocccus species 1) the presence of proteins involved in error free or in error-prone repair mechanisms (for the latter, mainly related to translesion polymerases activity (TLS) and accessory factors) and 2) two different regulation of DNA repair genes expression, one based on the well-known SOS regulators RecA and LexA, the second requiring two other proteins IrrE a metalloprotease able to cleave DdrO a transcriptional repressor of DNA repair genes.
This manuscript is well written, shows the evolution of knowledge in these topics both in Deinococcus species and in other model bacteria and pave the way for future studies. There are appropriate and adequate references to related works reviewed in this manuscript. Even if some results about DdrO and IrrE proteins were recently reviewed (Lim et al, 2019) by the same authors, very recent results as e.g. describing the molecular signaling mechanism that triggers DdrO cleavage have been included (Magerand et al, 2021) thus improving our knowledge in this topic.
I only have a few comments:
Lines 190-208: On the basis of their recent BLAST searches, the authors described homologs of TSL polymerases and of associated factors as well as one or two lexA-imuY-imuB-C-dnaE2 mutagenesis cassette(s) in several Deinococcus species but this work was not published or a reference is missing. If so,
- I suggest that the authors provide a table listing gene ID of these homologs found in these species and describe the settings they used to search homologs of these proteins.
- Does any polymerase V homolog was found in Deinococcus species?
- Finally, it remains a little bit unclear if only radioresistant or also several radiosensitive Deinococcus species encode these proteins.
Lines 209-221: The authors discussed the advantages and disadvantages of the coexistence in the same organism of the error- free and error-prone repair mechanisms, and their consequences on genetic variability but did not consider that:
- While D. radiodurans does not encode TSL polymerases, a mutagenesis system enhanced after irradiation, and based on transposition events has been described in this bacterium. Do translesion polymerases activities in D. deserti result in more mutants than transposition in D. radiodurans?
- D. radiodurans and D. deserti are polyploid cells containing several copies of each replicon. Can polyploidy be a brake or may enhance the mutagenesis induced by TSL activities?
